# Effect of Turning on the Surface Integrity and Fatigue Life of a TC11 Alloy in Very High Cycle Fatigue Regime

**Tao Gao [1], Zhidan Sun [2] , Hongqian Xue [1],\*, Emin Bayraktar [3],\* , Zhi Qin [1], Bin Li [1] and Han Zhang [1]**

[1]    School of Mechanical Engineering, Northwestern Polytechnical University, Xi'an 710072, China; gtao@mail.nwpu.edu.cn (T.G.); qinzhi@nwpu.edu.cn (Z.Q.); bin.li@mail.nwpu.edu.cn (B.L.); han.zhang@nwpu.edu.cn (H.Z.)
[2]    ICD, P2MN, LASMIS, University of technology of Troyes, CNRS FRE2019 Troyes, France; zhidan.sun@utt.fr
[3]    School of Mechanical and Manufacturing Engineering, Supmeca-Paris, 93407 Paris, France
\*    Correspondence: xuedang@nwpu.edu.cn (H.X.); bayraktar@supmeca.fr (E.B.); Tel.: +86-029-8849-2843 (H.X.); +33-676-103-620 (E.B.)

**Abstract:**    In this work, the effect of a turning process on fatigue performance of a Ti-6.5Al-3.5Mo-1.5Zr-0.3Si (TC11) titanium alloy is studied in the high cycle fatigue (HCF) and very high cycle fatigue (VHCF) regimes. For this purpose, the surface characteristics including surface morphology, surface roughness and residual stress were investigated. Moreover, axial fatigue tests were conducted with an ultrasonic fatigue testing system working at a frequency of 20 kHz. The results show that the turning process deteriorated the fatigue properties in both HCF and VHCF regimes. The fatigue strength at $1 \times 10^8$ cycles of turned samples is approximately 6% lower than that of electropolished ones. Fracture surface observations indicate that turning marks play a crucial role in the fatigue damage process, especially in the crack initiation stage. It was observed that the crack of all the turned samples originated from turning marks. In addition, the compressive residual stress induced by the turning process played a more effective role in resisting crack propagation in the VHCF regime than in the HCF regime.

**Keywords:** titanium alloy; turning surface integrity; fatigue strength; very high cycle fatigue; damage mechanism

## 1. Introduction

Titanium alloys are widely used in aero-engine components, airframe structures and landing gear parts due to their unique comprehensive performance including high strength, low density and excellent resistance to creep, corrosion and fatigue. The service life of key components, such as aero-engine compressor, gear and blades, are beyond $1 \times 10^7$ cycles, and thus in the very high cycle fatigue (VHCF) regime [1]. Since unexpected fatigue failures may still occur at the imposed stresses lower than the traditional fatigue limit, the traditional fatigue limit defined at $1 \times 10^7$ cycles is not suitable for the fatigue design of aero-engine components working in the VHCF regime. The VHCF properties are thus a vital indicator to assess the security of these aero-engine components. Therefore, the VHCF performance of titanium alloys and their damage mechanisms have received extensive attention over recent decades [2–4].

In the VHCF regime, crack initiation consumes more than 90% of total fatigue life in titanium alloys. Under VHCF, crack initiation tends to occur at the specimen subsurface and interior, instead of specimen surface in HCF and LCF regimes, and internal crack initiation area exhibits usually a fish-eye pattern [5–7]. The failure mechanisms in the VHCF regime are thus different from those in the HCF

and LCF regimes. Fractographic inspections revealed that facets are primary morphologies at the crack initiation area in most near-α and α + β titanium alloys in the VHCF regime [7–9], and fatigue crack initiation is ascribed to the formation of the facets. It has been reported that these facets are obtained by α grains fractured in a transcrystalline manner [7,8]. In order to reveal how the facets are formed, several studies have been conducted to characterize facet features such as the spatial distribution, crystallographic orientation and micro-morphology. Based on these studies, different formation mechanisms of various facets have been suggested [10,11]. Contrary to the presence of facet morphology, some studies revealed however that no facets can be observed at the surface and the internal crack initiation areas of α + β titanium alloys even in the VHCF regime [12,13]. The crack initiation in those cases is attributed to localized plastic slips or some silicide precipitates [4,13,14]. These different investigations can provide better knowledge of crack initiation and propagation mechanisms of titanium alloys in the VHCF regime. However, it should be noted that these investigations were mainly focused on the effect of inherent microstructure on VHCF damage, and little attention was paid to other effects, for example that of surface condition. Specimens used in these studies were usually carefully polished to avoid the effect of surface roughness, defects and residual stresses induced by a machining process. In fact, these defects are difficult to be avoided for machined parts, especially for titanium alloy components. Titanium alloys exhibit poor machinability due to low thermal conductivity, high chemical reactivity and low elastic modulus, which makes it difficult to control the machined surface integrity. The fatigue failure of aero-engine components often occurs from these surface machining defects in the VHCF regime [1]. It is well known that the surface integrity directly affects the fatigue life and the strength of machined components. It is thus important for the aviation engineering industry to investigate the effect of machining surface integrity of titanium alloys on the fatigue performance as well as the damage mechanisms in the VHCF regime.

Some research work has been conducted to investigate the effect of surface integrity on the fatigue behavior of metal alloys and components [15–18]. Among these different surface integrity characteristics, the fatigue life of machined specimens was mainly determined by the interaction of surface roughness, residual stress and work hardening layer [15]. It is well known that rough surfaces with groove traces, scratch marks and other defects are potential sites of fatigue crack initiation, and consequently decrease fatigue life. Zhang et al. [17] investigated the effect of creep feed grinding on fatigue performance of a $Ni_3Al$ alloy and showed that the surface roughness has a dominant deteriorative effect on the fatigue life because of stress concentration induced by processing defects. Nevertheless, surface compressive residual stress and the work hardening layer exhibit positive effects on fatigue performance by delaying fatigue crack initiation and propagation, which leads to prolonged fatigue life. Liu et al. [15] pointed out through their study that the deleterious effect of surface roughness on fatigue performance can be overshadowed by compressive residual stress for 17-4PH stainless steel. Javidi et al. [19] investigated the effect of surface integrity induced by turning on the fatigue behavior of a 34CrNiMo6 alloy and indicated that the influence of residual stress on fatigue life is more significant than the effect of surface roughness. Due to the complex interaction between surface integrity characteristics, such as surface roughness and residual stress induced by different machining processes, the effect of surface integrity on fatigue life and the corresponding damage mechanisms need to be further investigated, especially in the VHCF regime.

In this work, the effect of surface integrity obtained by turning on the fatigue performance of a Ti-6.5Al-3.5Mo-1.5Zr-0.3Si alloy (TC11) was investigated in the HCF and the VHCF regimes. The surface morphology, surface roughness and residual stress induced by the turning process were first characterized. Then, the fatigue lives of turned samples were obtained through high-frequency ultrasonic fatigue tests. The fatigue data and the fracture surfaces of turned samples were compared to those of electropolished specimens obtained in our previous work in order to reveal the effect of the turning process on fatigue life and the damage mechanisms of the TC11 titanium alloy, especially in the VHCF regime.

## 2. Materials and Methods

### 2.1. Material

A commercial Ti-6.5Al-3.5Mo-1.5Zr-0.3Si (TC11) titanium alloy was investigated in this work. It is an α + β titanium alloy usually used in aeronautical structures, such as aero-engine compressors, blades and turbine discs. This alloy has higher strength, superior creep resistance and better thermal stability in comparison with Ti-6Al-4V alloy thanks to the addition of silicon. The small amount of silicon can reduce dislocation mobility, which allows improving creep resistance and stabilizing the microstructure under moderate temperature to meet the service requirements of aero-engine components. After grinding and polishing, the microstructure of the studied TC11 alloy was observed by scanning electron microscope (SEM). Metallographic characteristics of transverse and longitudinal sections are shown in Figure 1a,b, respectively. The microstructure is composed of primary alpha phase ($\alpha_p$) and transformed β matrix ($\beta_t$), corresponding, respectively, to the dark and bright regions indicated in Figure 1. The volume fraction of $\alpha_p$ is about 60%.

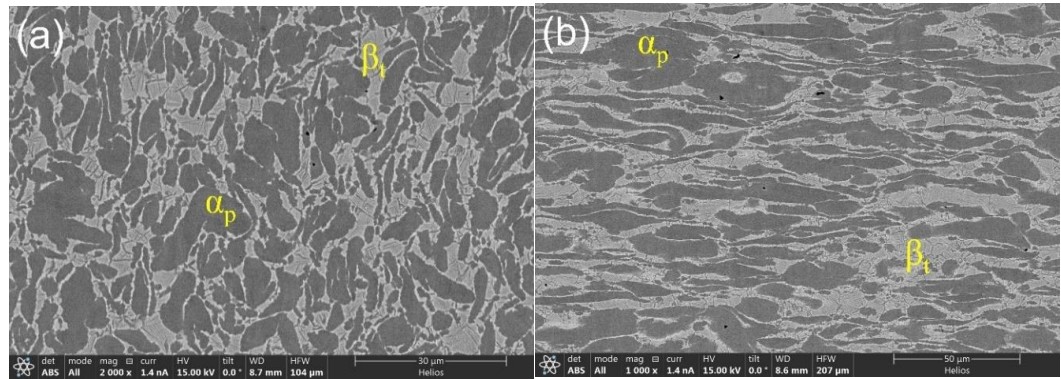

**Figure 1.** Microstructure images of the studied TC11 titanium alloy: (**a**) transverse section, and (**b**) longitudinal section.

### 2.2. Fatigue Sample and Surface Characteristics

In this work, hourglass-shaped fatigue sample was designed through analytical and finite element methods to meet the requirements of 20 kHz ultrasonic resonant vibration. The dimensions and the geometry of the sample are shown in Figure 2. The minimum diameter at the central reduced section is 4 mm. Fatigue samples were machined by turning and the processing parameters are cutting speed $v_c$ = 10 m/min, feed rate $f_z$ = 0.04 mm/rev and depth of cut $\alpha_p$ = 0.1 mm. In order to quantitatively analyze the influence of turning on fatigue performance of the TC11 alloy, some samples were polished using SiC paper and then electropolished in a solution composed of $HClO_4$ and ethanol to obtain a mirror-like surface and eliminate surface defects and residual stress induced by turning. The electropolished samples are considered as a reference to analyze the effect of surface integrity resulting from turning on the HCF and VHCF performance. The fatigue performance of electropolished samples of the TC11 alloy has been investigated in our previous work [13].

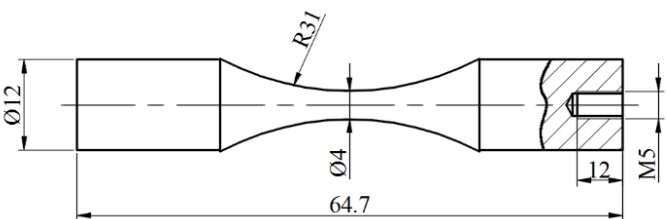

**Figure 2.** Geometry and dimensions of the fatigue sample (unit: mm).

In order to investigate the role of surface roughness and morphology in fatigue behaviour, surface roughness profiles in the central region of both electropolished and turned samples were measured using a contact profilometer (Mahr Marsurf M300, Carl-Mahr-Str., Göttingen, Germany). The values of typical surface roughness parameters were determined based on EN ISO 4287 and EN ISO 16610-21 standards. The surface morphologies of the electropolished and the turned samples were observed using SEM to characterize surface defects and microstructure.

Axial residual stress generated by the turning process was measured using an X-ray diffractometer (LXRD MG2000, Proto Manufacturing Ltd. 2175 Solar Crescent Oldcastle, ON, Canada) with Cu K$\alpha$ radiation and Bragg angle of 142° in the {213} plane. The voltage and current were 25 kV and 30 mA, respectively. The values of the residual stress were calculated using the $\sin^2 \psi$ method. To obtain the in-depth distribution of the residual stress, thin surface layers of the turned sample were removed successively by electrolytic polishing in a solution composed of $HClO_4$ and Ethanol.

### 2.3. Fatigue Tests and Fracture Surface Observation

Axial tension-compression fatigue tests were conducted using an ultrasonic fatigue testing system with a frequency of 20 kHz and a stress ratio of $R = -1$. A pulse-pause loading mode and a compressed air cooler were used to prevent the temperature rising of samples generated by high-frequency cyclic loading. The temperature in the centre of tested samples was controlled to be below 30 °C. The applied stress amplitudes were chosen in the range of 640–740 MPa so that the fatigue lives are in the HCF and VHCF regimes. After fatigue tests, the fracture surfaces of the failed samples were observed using SEM to analyze the failure mechanisms.

## 3. Results

### 3.1. Surface Roughness and Morphology

The surface profiles of the turned and the electropolished samples are illustrated in Figure 3. For the turned sample, there is the presence of a regular fluctuating profile, and the maximum profile height induced by turning is about 2.5 µm (Figure 3a), which is much larger than that of the electropolished sample. The electropolished sample has a much smoother surface and the maximum profile height is blow 0.5 µm (Figure 3b), which means that the electropolishing process effectively reduced the surface roughness and improved the surface quality. Several typical values of surface roughness, such as arithmetic mean roughness $R_a$, mean roughness depth $R_z$, difference in height between the highest peak and the deepest valley $R_t$, and mean peak width $R_{sm}$, were calculated according to the standards of EN ISO 4287 and EN ISO 16610-21, and the results are presented in Table 1.

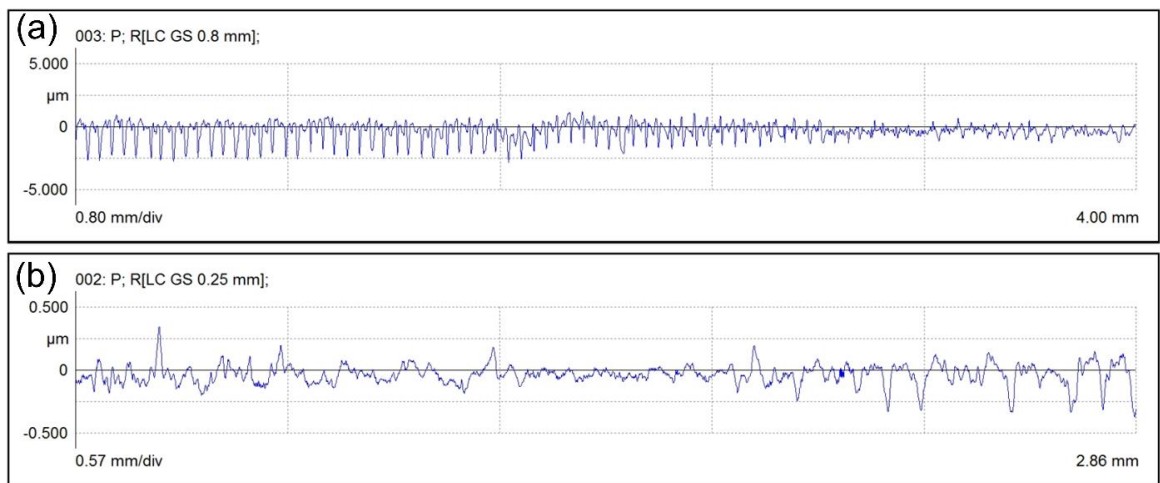

**Figure 3.** Surface profiles of (**a**) turned and (**b**) electropolished specimens.

**Table 1.** Values of several surface roughness parameters of the turned and the electropolished specimens.

| Treatments | $R_a$ (µm) | $R_z$ (µm) | $R_t$ (µm) | $R_{sm}$ (µm) |
|---|---|---|---|---|
| Turning | 0.51 | 3.11 | 4.03 | 46.56 |
| Electropolishing | 0.06 | 0.43 | 0.72 | 101.12 |

Considering that surface profile and roughness parameters are not sufficient to describe surface topographical features, the surface morphologies of the electropolished and the turned samples were observed using SEM and they are shown in Figure 4. It can be seen that the surface of the electropolished sample presents free-defects characteristics (Figure 4a,b), and is similar to the microstructure features highlighted at the longitudinal section shown in Figure 1b. These electropolished samples with high surface quality would reveal the intrinsic damage mechanisms of the alloy itself and avoid the effect of turning on the damaging process. Different from the smooth surface of the electropolished sample, the surface obtained by turning is obviously rough, as shown in Figure 4c,d. A series of defects such as grooves, fold and scratch markings can be observed at the turning surface. These surface turning defects could give rise to stress concentration and promote fatigue crack initiation. Similar turning defects were also observed by several researchers for titanium alloys [20–22]. It has been suggested that the grooves result from the mapping of microchipping of tool cutting edges to the machined surface [22,23], which is related to the plowing effect of the built-up edges [24,25].

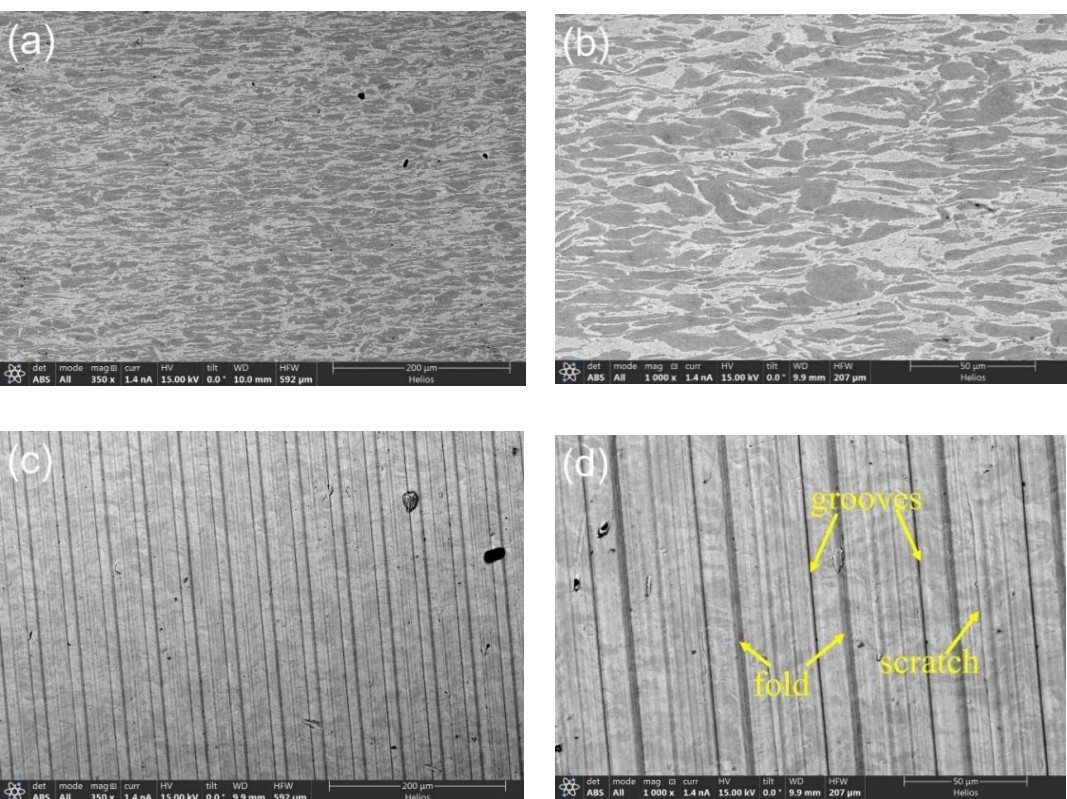

**Figure 4.** SEM surface morphology observations of (**a**,**b**) electropolished and (**c**,**d**) turned samples.

*3.2. Residual Stress Measurement*

The in-depth axial residual stress profile of the turned sample is shown in Figure 5. Considering the residual stress relaxation induced by iterative removal of material, the measured values of residual stress were corrected using a mathematical method proposed by Moore and Evans [26]. The depth of the compressive residual stress field generated by the turning is about 30 µm. The maximum compressive residual stress is located at the turned surface and the corresponding value is about −420 MPa.

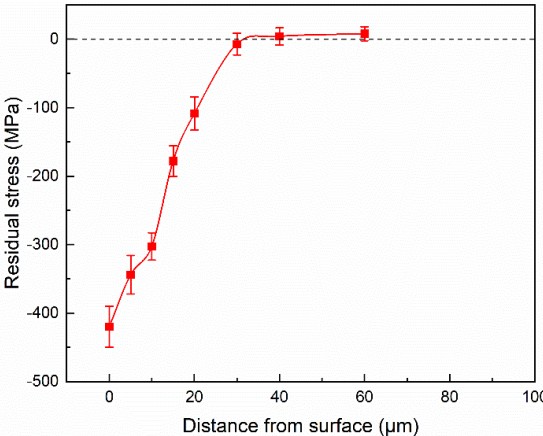

**Figure 5.** In-depth residual stress of a turned sample.

### 3.3. Fatigue Life Analysis

The fatigue test results of both electropolished and turned samples of the TC11 alloy are presented in the form of an *S-N* plot, as shown in Figure 6. It should be noted that the fatigue life data and the fracture surface observation of electropolished samples were obtained in our previous work, and they are presented in this paper as a reference to analyze the effect of turning on fatigue life and damage mechanism. The fatigue lives of the electropolished and the turned samples exhibit different tendencies depending on the applied stress amplitudes. The fatigue life of the turned samples continuously increases with the decrease in applied stress and presents a lower scatter in comparison with the electropolished samples. It seems that there is no transition between the HCF regime and the VHCF regime for the fatigue data obtained with the turned samples. The fatigue datapoints obtained for the turned samples can be fitted using an *S-N* curve which was determined by a linear regression model according to the ASTM standard [27]. However, the *S-N* curve of the electropolished samples exhibits multi parts with different slopes between the two regimes. The SEM observations of the fracture surfaces indicate that fatigue failure of all the turned samples occurred at sample surface, whereas subsurface-induced or interior-induced failure occurred in the VHCF regime for the electropolished samples. The difference in fatigue life distribution between electropolished and turned samples could be ascribed to the different damage processes, which will be discussed in the following sections.

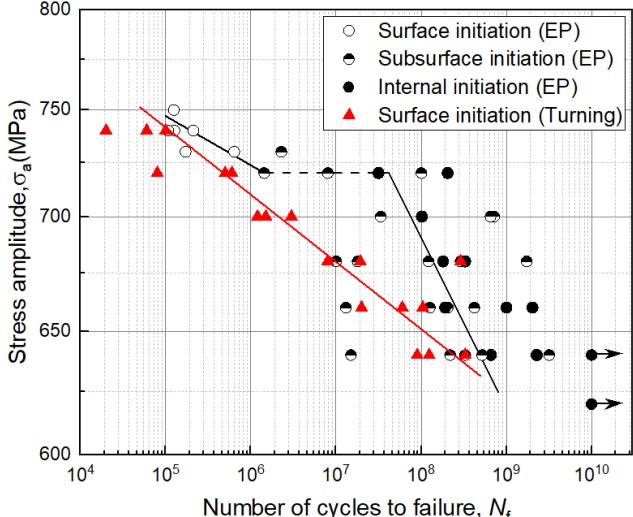

**Figure 6.** S-N plot for the fatigue test data of electropolished (EP) and turned samples for the studied TC11 alloy.

As shown in Figure 6, the obtained fatigue lives of turned samples vary from about $2 \times 10^4$ cycles to $3.29 \times 10^8$ cycles under the applied stress amplitudes ranging from 740 to 640 MPa. The limited fatigue life data of turned and electropolished samples seem to show that the turned samples exhibit lower fatigue resistance than the electropolished ones in both HCF and VHCF regimes, especially in the fatigue life range of $1 \times 10^6$–$2 \times 10^8$ cycles. The fatigue strengths of turned and electropolished samples at $10^8$ cycles ($\sigma_{(10^8)}$) were calculated based on a failure probability of 50%. The values of fatigue strength at $10^8$ cycles for the electropolished and the turned samples are 690 and 651 MPa, respectively. The fatigue strength $\sigma_{(10^8)}$ of the turned samples is approximately 6% lower than that of the electropolished ones. For the turned samples, the synergetic effect between applied stress amplitude and surface characteristics is responsible for the reduced fatigue strength.

### 3.4. Fracture Surface Observation

The fracture surfaces of both electropolished and turned samples were observed using SEM to reveal the effect of turning surface on the fatigue properties of the studied TC11 alloy. Considering that the crack initiation and early propagation process consumes the majority of fatigue life in HCF and VHCF regimes, SEM observation of fracture surfaces was mainly focused on the crack initiation and early propagation areas.

Fracture surface observations indicate that for the electropolished samples, fatigue crack initiated from surface in the HCF regime, whereas it initiated from the subsurface or interior in the VHCF regime. Two representative fracture surfaces of electropolished samples failed at $\sigma_a = 730$ MPa with $N_f = 6.39 \times 10^5$ in the HCF regime and at $\sigma_a = 680$ MPa with $N_f = 1.81 \times 10^8$ in the VHCF regime are presented in Figures 7 and 8, respectively. Under high applied stress amplitudes, the electropolished samples failed from surface. The crack initiation source started from a small surface area, as shown in Figure 7a–c. The area near the surface crack initiation site exhibits a rough appearance with the presence of dimples, grooves and particles, but without facets, as shown in Figure 7c,d. It was revealed in our previous work [13] that these particles in the crack initiation region are silicides. The crack initiation in the HCF regime is attributed to local stress concentration induced by large silicides. Under low applied stress amplitudes, the fatigue crack of electropolished samples initiated from subsurface or interior. The internal crack initiation area exhibits a fish-eye pattern, as shown in Figure 8. It is generally reported that in the VHCF regime, the internal crack initiation of $\alpha + \beta$ titanium alloys is ascribed to facet clusters of $\alpha_p$ grains [8,9], and thus the fracture surface commonly presents brittle characteristics. However, in the present study, the internal crack initiation area with fish-eye pattern is covered by dimples, grooves and peak-like features instead of facets, as shown in Figure 8c,d. This means that the fracture surface of this TC11 alloy exhibits typical ductile damage appearance. At ultra-high resolution SEM, several nano-silicides were observed in the dimples at the fish-eye region, as shown in Figure 8d. In this case, the internal crack initiation is attributed to microvoids nucleation around nano-silicides.

Typical fracture surfaces of turned samples failed in the HCF and the VHCF regimes are presented in Figure 9. The fracture surface morphology and the crack initiation site for turned samples are different from those of electropolished ones. The crack initiation sites of turned samples are all located at the surface, and no internal crack initiation was observed even in the VHCF regime. With the decrease in applied stress, the crack initiation region extends to a larger area along the sample surface and the corresponding arc length of crack sources becomes larger. To highlight the difference in the microscopic topography of the fracture surface induced by the turning process, more detailed SEM observations of the surface initiation area were conducted and the representative morphologies of turned samples that failed in both HCF and VHCF regimes are shown in Figures 10 and 11.

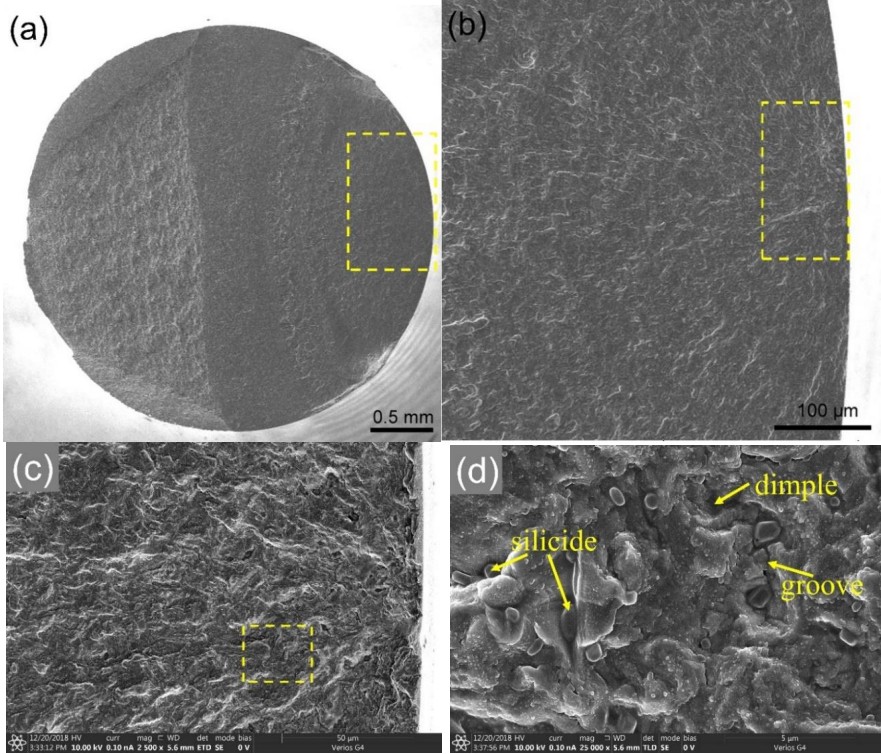

**Figure 7.** Fracture surface of an electropolished sample tested at $\sigma_a = 730$ MPa, failed after $N_f = 6.39 \times 10^5$ cycles in the HCF regime: (**a**) overall view showing the surface crack initiation, (**b**) magnified image of the rectangular area in (**a**), (**c**) magnified image of the rectangular area in (**b**) showing the surface crack initiation morphology, (**d**) magnified image of the rectangular area in (**c**) revealing the morphology of the area near the crack initiation site.

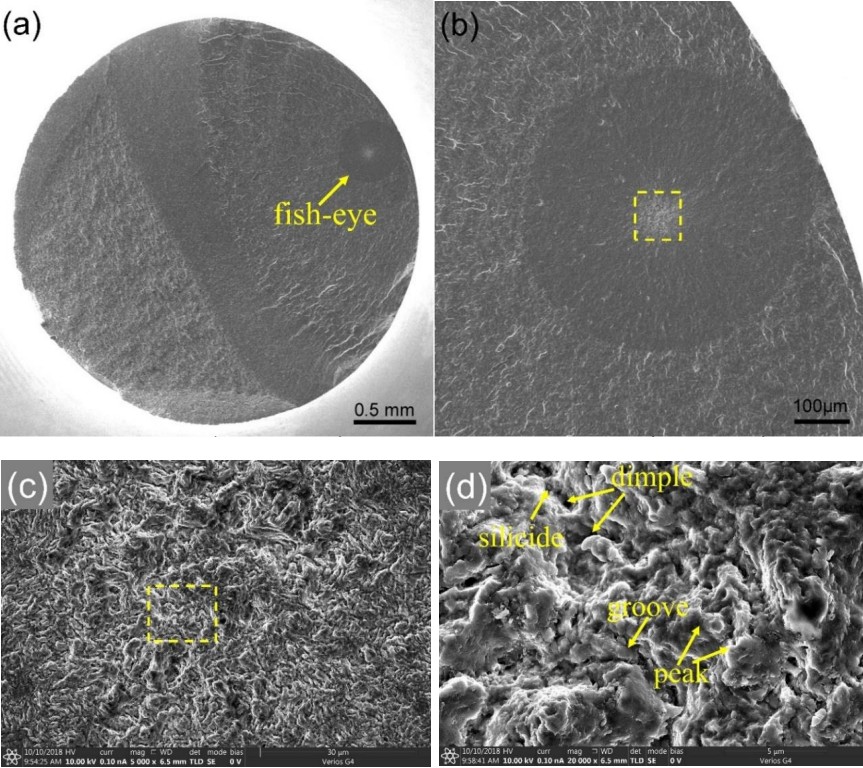

**Figure 8.** Fracture surface of an electropolished sample tested at $\sigma_a = 680$ MPa, failed after $N_f = 1.81 \times 10^8$

cycles in the VHCF regime: (**a**) overall view showing the internal crack initiation, (**b**) magnified image of the fish-eye pattern in (**a**), (**c**) central area of the fish-eye pattern in (**b**) showing ductile damage appearance, (**d**) detailed morphology of the rectangular area in (**c**) showing dimples, silicides, grooves and peaks.

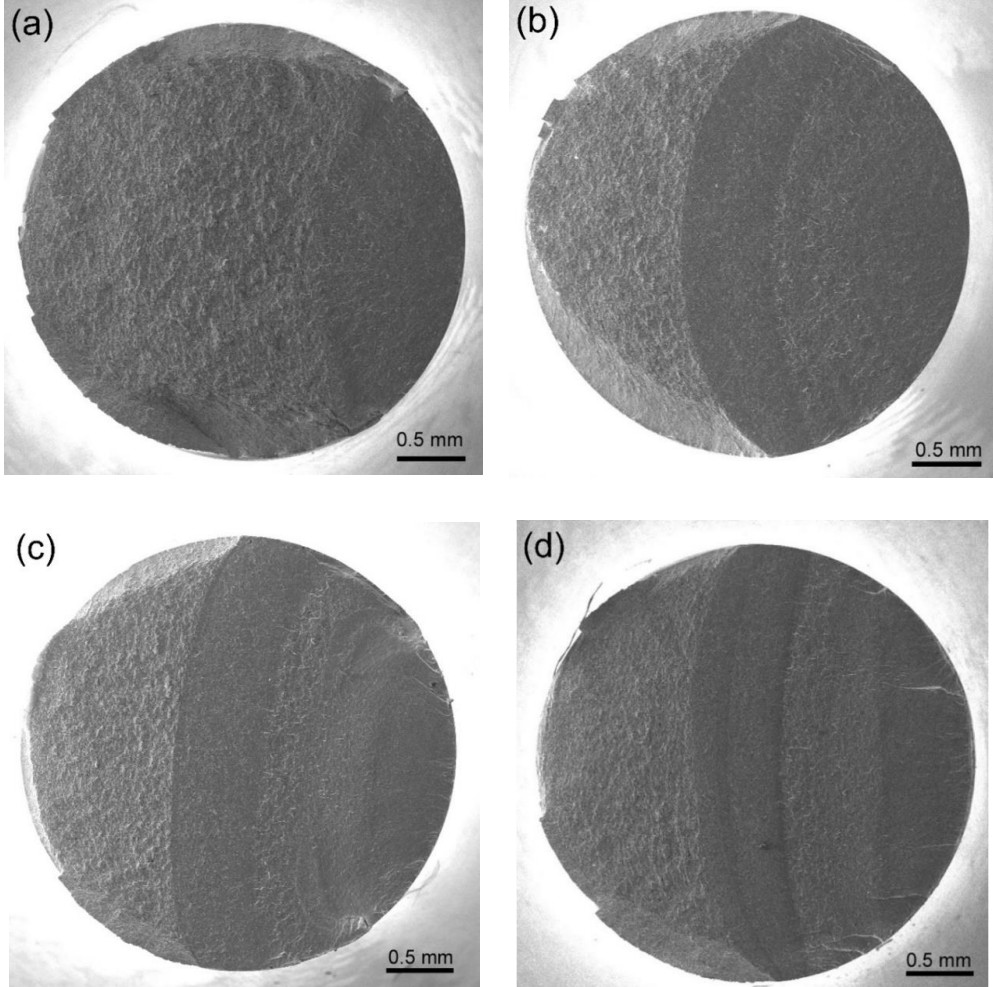

**Figure 9.** Typical macroscopic fracture surfaces of turned specimens failed at different fatigue lives: (**a**) $\sigma_a$ = 740 MPa, $N_f$ = 1 × 10$^5$, (**b**) $\sigma_a$ = 700 MPa, $N_f$ = 1.5 × 10$^6$ cycles, (**c**) $\sigma_a$ = 660 MPa, $N_f$ = 6 × 10$^7$ cycles and (**d**) $\sigma_a$ = 640 MPa, $N_f$ = 3.29 × 10$^8$ cycles.

Figure 10 shows the fracture surface and the morphology of cracks on the surface of a sample tested at $\sigma_a$ = 700 MPa and failed with a number of cycles to failure of $N_f$ = 1.5 × 10$^6$ cycles in the HCF regime. The fatigue crack initiation occurred at surface turning marks, as shown in Figure 10a,b. The arc length occupied by the crack initiation area at the sample surface is about 0.6 mm. No apparent dimples, grooves and silicide particles were observed in the crack initiation region. Friction-induced markings are the main microscopic appearance of the crack initiation area of turned samples failed in the HCF regime, as shown in Figure 10b. The formation of friction-induced markings may be attributed to contact between the two separated surfaces due to repeated tension-compression loading. From the surface morphology shown in Figure 10c, no secondary cracks along turning marks below the crack initiation site can be observed. It could be inferred that the HCF crack initiation occurred at a specific turning mark. Once a surface fatigue crack initiated, it propagated preferably along a specific turning mark and formed a quite flat surface crack path, as shown in Figure 10d.

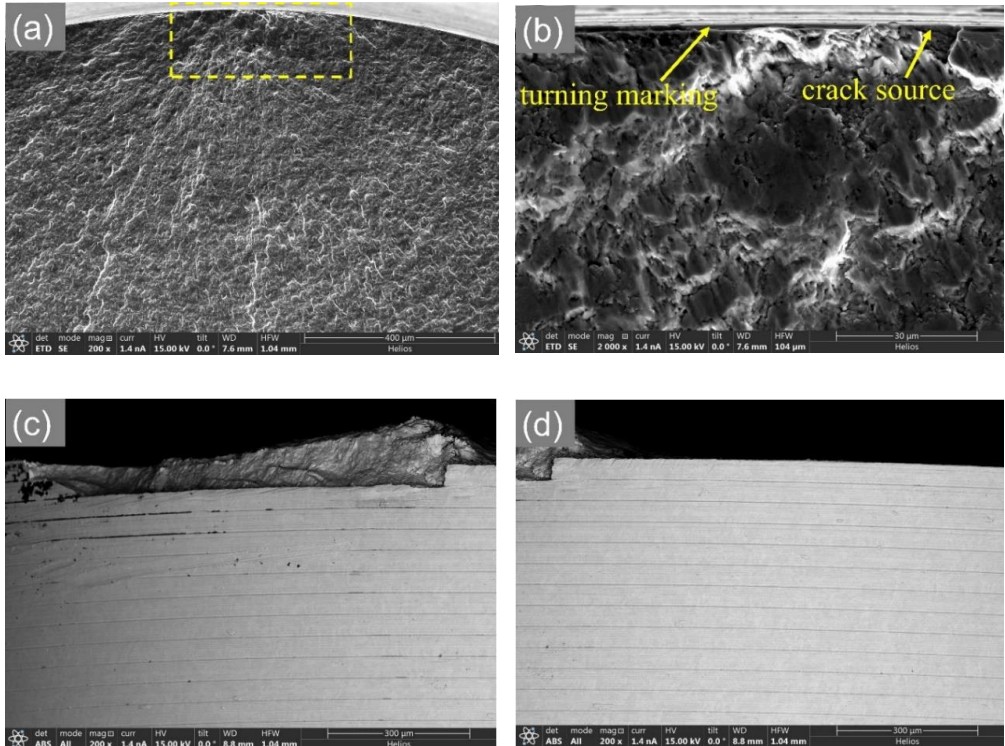

**Figure 10.** Fracture surface and surface morphology of a turned sample tested at $\sigma_a$ = 700 MPa and failed with $N_f$ = 1.5 × 10$^6$ in the HCF regime: (**a**) fracture surface showing surface crack initiation, (**b**) magnified image of the rectangular area in (**a**) revealing that the crack initiated from a turning mark, (**c,d**) morphological features near the crack initiation site on the turned surface revealing that the crack seemingly initiated from a specific turning mark and propagated rapidly along another turning mark.

Figure 11 shows the fracture surface of a sample tested at $\sigma_a$ = 640 MPa and failed after $N_f$ = 3.29 × 10$^8$ cycles in the VHCF regime. The VHCF crack also initiated from turning marks, but the fracture surface morphology exhibits a significant difference with respect to that of the sample failed in the HCF regime. It seems that the VHCF crack initiation resulted from several turning marks, instead of one single specific turning mark in the HCF regime, as shown in Figure 11a,d,e. The corresponding arc length of surface crack initiation area increases obviously and the value is more than 4 mm. A bright band boundary (indicated by a white arrow in Figure 11a) near the sample surface and several tear ridges can be observed at the macroscopic fracture surface. At high magnification, the bright band boundary shows more likely tearing steps, as shown in Figure 11b,c. The regions on both sides of the boundary show completely different fracture morphologies. For the convenience of presentation, the area outside and inside the bright band boundary is defined as Region I and Region II, respectively, as shown in Figure 11b. The thickness of Region I is in a range of 20–25 μm, which is consistent with the depth of the compressive residual stress layer (Figure 5). Several step cracks and numerous microcracks can be observed in Region I. These step cracks are attributed to the junction of adjacent surface cracks which initiated from different surface turning marks. The direction of intensive microcracks is approximately parallel to the turning marks and perpendicular to the crack propagation direction. The presence of microcracks in Region I can be attributed to the slow growth of surface cracks in the layer where the compressive residual stress is present. The bright band boundary indicates that the surface crack tips did not continuously propagate inward, and it can be inferred that the surface crack tips stopped propagating temporarily at the bright band boundary. When adjacent surface cracks both propagated at the bright band boundary, they coalesced to form a larger surface crack and then continued to propagate until failure.

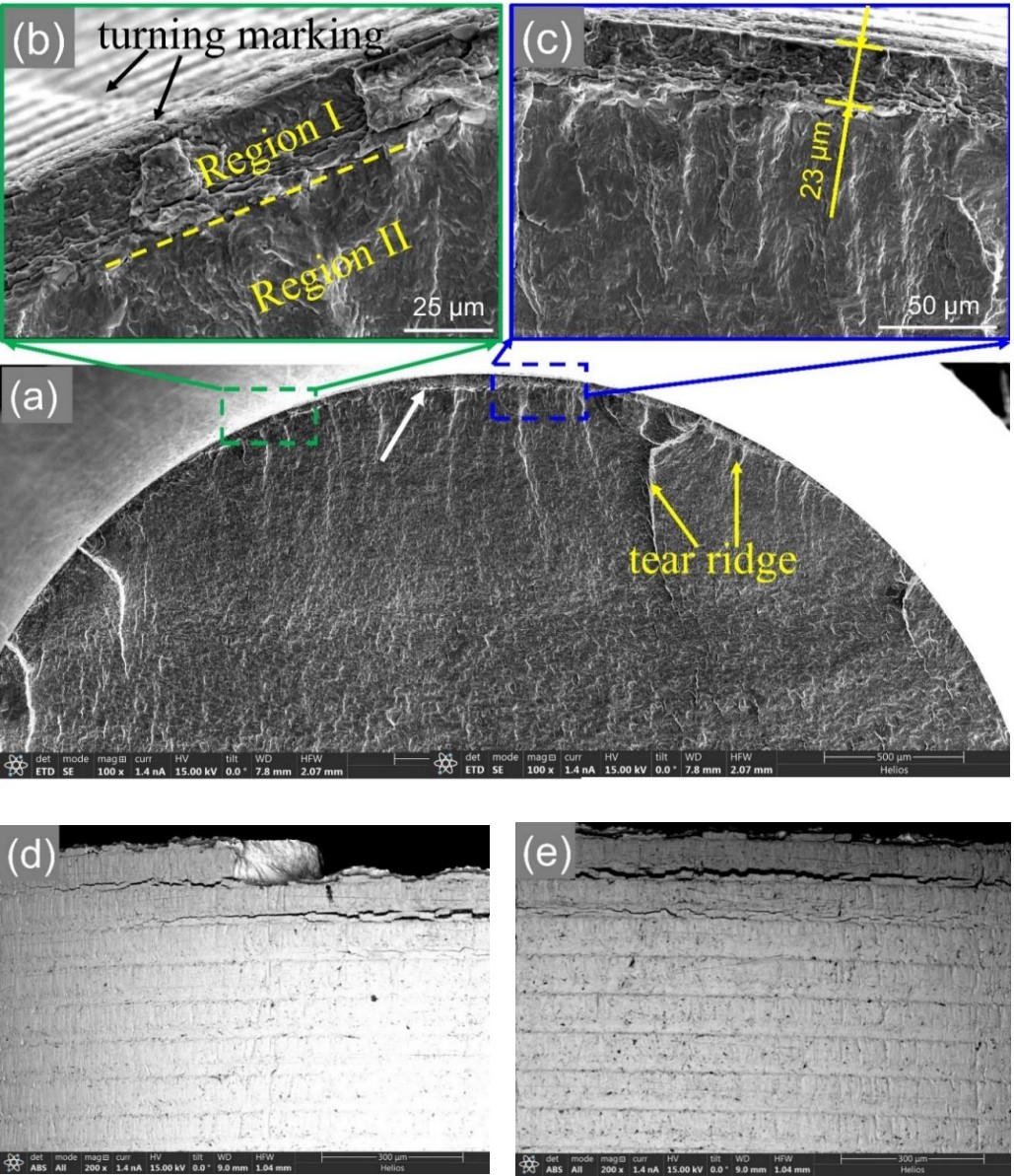

**Figure 11.** Fracture surface and surface morphology of a turned sample tested at $\sigma_a$ = 640 MPa and failed after $N_f$ = 3.29 × 10$^8$ cycles in the VHCF regime: (**a**) fracture morphology of the surface crack initiation area, (**b,c**) magnified image of the local crack feature on the surface, (**d,e**) several cracks along turning marks below the crack initiation site on the turned sample surface.

The VHCF crack initiation and propagation scenario described above depends on the applied stress amplitude and the sample state including surface roughness, turning marks and residual stress induced by the turning process. These elements will be discussed in the next section.

## 4. Discussion

The microstructure played a determining role in fatigue crack initiation for electropolished samples in both HCF and VHCF regimes. Silicides are the main micro-defects affecting the fatigue properties of the TC11 alloy. The fatigue crack of electropolished samples initiated from large micro-silicide particles or clusters, as shown in Figures 7d and 8d. The crack initiation site and the fracture surface morphology are closely related to the morphology of silicides and the applied stress amplitude.

For the turned samples, the surface integrity features overcame the effect of silicides on fatigue cracking and dominated the crack initiation process. All the cracks of turned samples initiated from

surface without exception, even in the VHCF regime, as shown in Figures 10 and 11. The difference in fracture morphology and fatigue cracking process between turned samples and electropolished ones are attributed to the surface integrity.

Among different surface integrity indicators, it has been reported that the fatigue properties of machined samples mainly depend on complex interaction between surface roughness and residual stress [15]. Surface roughness and defects are treated as a deleterious factor, as they generate local stress concentration, which induces surface crack nucleation and thus decreases the fatigue properties [28]. The SEM observation of the fracture surface revealed the multiple crack initiation sites on the turned sample surface. These multiple crack initiation sites are typical characteristics of notched samples. The rough surface and defects can be considered as micro-notch, and the negative effect on fatigue performance can be quantified by geometrical stress concentration factor $K_t$. It has been proposed that the stress concentration factor of rough surface can be determined by the relation $K_t = 1 + 4.0 (R_t/R_{sm})^{1.3}$ [29]. The stress concentration factor of the turned rough surface obtained using the relation is 1.16. The steep stress gradient resulting from the notch effect provides more critical sites for crack initiation.

As regards the influence of residual stress, it is clear that compressive residual stresses can delay crack initiation and propagation, which can reduce or compensate the negative effects of rough surface on fatigue properties. It has been reported that a deep compressive residual stress belt with large values can prevent crack initiation from occurring at sample surface and can consequently improve fatigue strength [30]. In the present study, all the cracks initiated from turned surface due to a high stress concentration factor generated by deeper turning marks. In the HCF regime, a high applied stress can promote the crack to initiate from a turning mark, as illustrated in Figure 12a,b. The compressive residual stress is not high enough to prevent the crack initiation from occurring at deep turning marks where the stress is strongly concentrated, but can only delay the crack initiation and early propagation. Therefore, the fatigue crack propagated continuously from the surface initiation sites until the final fracture, as shown in Figure 12c. However, the role played by compressive residual stress to delay crack initiation and early propagation seems to be more obvious in the VHCF regime. This is because more extended surface crack initiation and an early propagation area can be observed in the annular belt with compressive residual stress. The thickness of the annular crack initiation and early propagation zone (Region I, as shown in Figure 11b,c) is in a range of 20–25 μm, which is consistent with the depth of the compressive residual stress layer. Due to the effect of compressive residual stress on delaying crack initiation and early propagation, it is difficult for initial surface cracks to coalesce to form/join the main crack. As fatigue damage accumulated, several cracks initiated from turning marks, as shown in Figure 12d,e. When the cracks' tips were in the turning affected region, these surface cracks propagated slowly along the depth direction and simultaneously tended to propagate along the surface direction. These cracks coalesced, which gave rise to the formation of an annular crack initiation and propagation area. Once these cracks broke through the compressive residual stress layer, they would further coalesce and propagate more rapidly along the depth direction until the final failure, as described in Figure 12f.

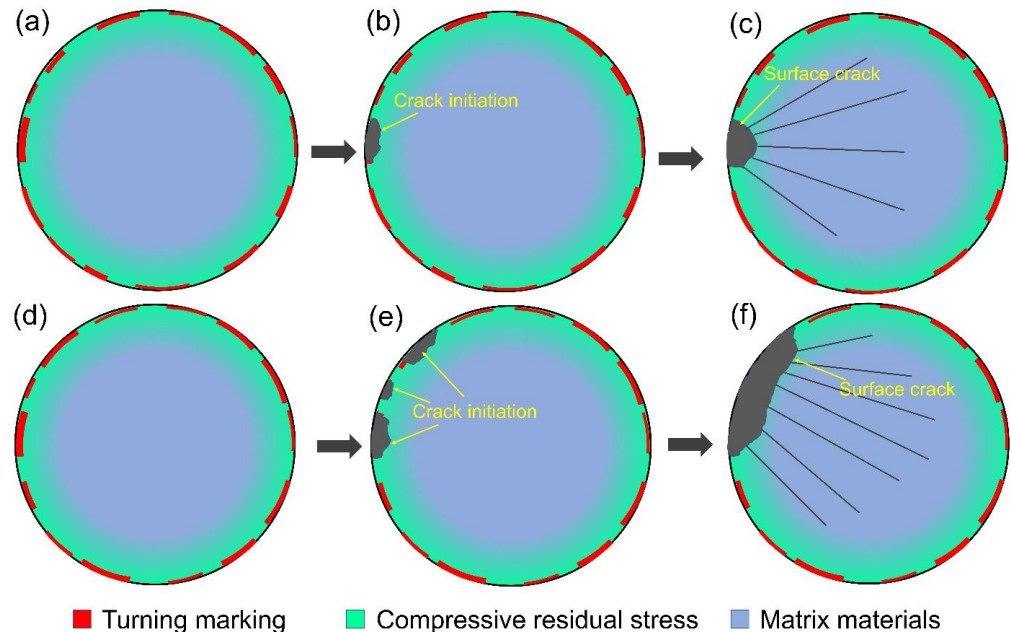

**Figure 12.** Schematic description of crack initiation and early propagation of turned samples at high applied stress in the HCF regime (**a–c**), and at low applied stress in the VHCF regime (**d–f**): (**a**) and (**d**) surface turning marks and compressive residual stress induced by the turning process, (**b**) crack initiating from a deep surface turning mark at high applied stress, (**c**) surface crack breaking through the compressive residual stress belt and rapid propagation until final fracture, (**e**) several cracks initiating from turning marks and tending to coalesce along the surface direction at low applied stress, (**f**) these cracks slowly propagating and coalescing to an annular notch in the compressive residual stress belt, and then rapidly propagating until final fracture.

## 5. Conclusions

In this work, the effect of turning surface integrity on fatigue performance of a Ti-6.5Al-3.5Mo-1.5Zr-0.3Si alloy was investigated using an ultrasonic fatigue testing system. The surface integrity parameters including surface morphology, surface roughness and residual stress were characterized. SEM fracture surface observations of electropolished and turned samples were conducted to reveal the difference in damage mechanisms between the electropolished and the turned samples. Some primary conclusions can be drawn as follows:

(1) The turning process generates rough surface with a series of grooves, fold and scratch marks. The maximum compressive residual stress induced by turning is located at the sample surface and the value is about −420 MPa. The depth of the compressive residual stress field of turned samples is about 30 μm;

(2) The turning surface has a deteriorating effect on the fatigue properties of the studied TC11 alloy in the VHCF regime, especially in the fatigue life range of $1 \times 10^6$–$2 \times 10^8$ cycles. The fatigue strengths $\sigma_{(10^8)}$ of turned samples is approximately 6% lower than that of electropolished ones in the VHCF regime. From an engineering perspective, more attention should be paid to the turning deteriorating effect on the fatigue performance of the titanium components, which are subjected to fatigue loading in the range of $1 \times 10^6$–$2 \times 10^8$ cycles in their working service.

(3) Turning marks play a dominant role in the fatigue damage process. All cracks of turned samples initiated from turning marks and no internal crack initiation was observed for the turned samples in both HCF and VHCF regimes. Optimizing turning process and decreasing deep turning marks is the key to improving the HCF and VHCF properties of titanium components;

(4) Under high applied stresses, the cracks initiated from a single deep turning mark and then propagated continuously until the final fracture. On the contrary, under low applied stresses,

cracks initiated from several turning marks and coalesced to form an annular crack initiation and early propagation area, roughly in the compressive residual stress belt. The compressive residual stress played a more effective role in resisting crack propagation in the VHCF regime than in the HCF regime.

**Author Contributions:** T.G. investigation, data curation, writing—original draft; Z.S. writing—review and editing; H.X. and E.B. conceptualization, methodology, supervision; Z.Q., B.L. and H.Z. validation, data curation. All authors have read and agreed to the published version of the manuscript.

**Funding:** This work was funded by the National Natural Science Foundation of China (91860206) and Shaanxi Province Key Research and Development Program (2019KW-063).

**Acknowledgments:** We would like to thank the Analytical & Testing Center of Northwestern Polytechnical University for SEM observations.

**Conflicts of Interest:** The authors declare no conflict of interest.

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
