# Peer review of "Effect of Turning on the Surface Integrity and Fatigue Life of a TC11 Alloy in Very High Cycle Fatigue Regime"

_metals, doi:10.3390/met10111507_

Round 1

Reviewer 1 Report

The main question addressed by the research is comparison of turning and electropolishing regarding their effects on crack initiation/propagation and fatigue life in HCF and VHCF regimes.

From my knowledge the work is original. The authors properly cite their previous work. The paper is well written, original and understandable. The conclusions are consistent with the evidence and arguments presented. They address the main question posed.

Author Response

Dear Reviewer,

Thank you for your suggestions and your contribution

We have profited from this occasion we have revised and imrproved our text

Kind Regards

E. BAYRAKTAR

Reviewer 2 Report

Interesting contribution.

1.In Figure 3. the scale should be in the same order (to allow easier comparison).

2.The author should regard the work by Denkena/Poll et al. on influence of residual stresses by deep rolling on fatigue life (compare Pape, F., Neubauer, T., Maiß, O., Denkena, B., Poll, G. (2017): Influence of Residual Stresses Introduced by Manufacturing Processes on Bearing Endurance Time, Tribology Letters, S. 65 – 70, DOI: 10.1007/s11249-017-0855-3) 

Author Response

Dear Reviewer,

Thank you for your comment and contribution

We thank you so much for improving our paper

For your valuable contribution.

  1. In Figure 3. the scale should be in the same order (to allow easier comparison).

Response: The profile height of sample surface processed by turning is about 2.5 μm, while the one obtained by electropolishing is approximate 0.25 μm. There are significant differences in the height of surface profile for the above two kinds of samples. The surface profiles, therefore, are presented in different scale in Figure 3a and b respectively, so as to identify surface profiles clearly.

The detailed comparison of surface roughness parameters is presented in Table 1.

2.The author should regard the work by Denkena/Poll et al. on influence of residual stresses by deep rolling on fatigue life (compare Pape, F., Neubauer, T., Maiß, O., Denkena, B., Poll, G. (2017): Influence of Residual Stresses Introduced by Manufacturing Processes on Bearing Endurance Time, Tribology Letters, S. 65 – 70, DOI: 10.1007/s11249-017-0855-3) 

Response:

Thank you for your comments. The work presented by Denkena/Poll et al. explored the effect of residual stress induced by different manufacturing processes on the bearing fatigue properties using experimental and finite element methods. The work can offer new insight into the effect of manufacturing process on fatigue properties of industrial parts. Therefore, we cited the reference to supplement the background in the present work, which is highlighted in red color in Section 1 (Page 2) and References (Page 16).

Reviewer 3 Report

The topic of the post is original. Authors from various workplaces paid special attention to the scientific processing of the paper. The article has the correct logical structure. In the introduction, the authors explained the motivation for solving the problem. The use of steel is mainly in the aerospace industry. This places special demands on the quality of the material.

The authors performed many experimental tests, and processed the obtained data by scientific methods. The core of the paper is a description of materials and methods for achieving results. The methods used are current and correct.

In conclusion, the authors specify the knowledge gained by scientific research of specified samples. Quantitative and qualitative parameters of the examined samples under specific test conditions are determined here. I recommend supplementing the conclusion by expressing the authors' opinions on the properties of the material with regard to its use. Assessment of the influence of properties on real use in practice. The conclusion can be supplemented by recommendations of the authors to improve the situation, or recommendations to limit the use in specific situations.

Author Response

The topic of the post is original. Authors from various workplaces paid special attention to the scientific processing of the paper. The article has the correct logical structure. In the introduction, the authors explained the motivation for solving the problem. The use of steel is mainly in the aerospace industry. This places special demands on the quality of the material.

The authors performed many experimental tests, and processed the obtained data by scientific methods. The core of the paper is a description of materials and methods for achieving results. The methods used are current and correct.

In conclusion, the authors specify the knowledge gained by scientific research of specified samples. Quantitative and qualitative parameters of the examined samples under specific test conditions are determined here. I recommend supplementing the conclusion by expressing the authors' opinions on the properties of the material with regard to its use. Assessment of the influence of properties on real use in practice. The conclusion can be supplemented by recommendations of the authors to improve the situation, or recommendations to limit the use in specific situations.

RESPONSE:

Thank you for your comments. From an engineering perspective, several suggestions have been supplemented the conclusion, as following (Page 14):

(2)  The turning surface has a deteriorating effect on the fatigue properties of the studied TC11 alloy in the VHCF regime, especially in the fatigue life range of 1×106 - 2×108 cycles. The fatigue strengths σ(108) of turned samples is approximately 6% lower than that of electropolished  ones in the VHCF regime. From an engineering perspective, it should be paid more attention to the turning deteriorating effect on the fatigue performance of titanium components, which are subjected to fatigue loading in the range of 1×106 - 2×108 cycles in their working service.

(3)  Turning marks play a dominant role in the fatigue damage process. All cracks of turned samples initiated from turning marks and no internal crack initiation was observed for the turned samples in both HCF and VHCF regimes. Optimizing turning process and decreasing deep turning marks is the key to improve HCF and VHCF properties of titanium components.

Reviewer 4 Report

    Dear authors, Although this is not state-of-the-art research, it has undoubted informational value. I do not contemplate any essential changes to the original manuscript

Author Response

Dear authors, Although this is not state-of-the-art research, it has undoubted informational value. I do not contemplate any essential changes to the original manuscript.

Dear Reviewer

Thank you so much for your valuable time to evaluate our manuscript

Thank you so much

Kind Regards

E. Bayraktar
